**Data Availability Statement:** Data can be obtained at www.measuredhs.com and is accessible online.

# The effect of dietary diversity on anemia levels among children 6–23 months in sub-Saharan Africa: A multilevel ordinal logistic regression model

Abdu Hailu Shibeshi[1]*, Kusse Urmale Mare[2], Bizunesh Fantahun Kase[3], Betel Zelalem Wubshet[2], Tsion Mulat Tebeje[4], Yordanos Sisay Asgedom[5], Zufan Alamrie Asmare[6], Hiwot Altaye Asebe[3], Afework Alemu Lombebo[7], Kebede Gemeda Sabo[2], Bezawit Melak Fente[8], Beminate Lemma Seifu[3]

1 Department of Statistics, College of Natural and Computational Science, Samara University, Samara, Ethiopia, 2 Department of Nursing, College of Medicine and Health Sciences, Samara University, Samara, Ethiopia, 3 Department of Public Health, College of Medicine and Health Sciences, Samara University, Samara, Ethiopia, 4 School of Public Health, College of Health Sciences and Medicine, Dilla University, Dilla, Ethiopia, 5 Department of Epidemiology and Biostatics, College of Health Sciences and Medicine, Wolaita Sodo University, Soddo, Ethiopia, 6 Department of Ophthalmology, School of Medicine and Health Science, Debre Tabor University, Debre Tabor, Ethiopia, 7 School of Medicine, College of Health Science and Medicine, Wolaita Sodo University, Soddo, Ethiopia, 8 Department of General Midwifery, School of Midwifery, College of Medicine and Health Science, University of Gondar, Gondar, Ethiopia

* abduhailu01@gmail.com

## Abstract

### Background

Anemia is the most common hematologic disorder of children worldwide. Since dietary diversity is a main requirement of children is to get all the essential nutrients, it can thus use as one of the basic indicator when assessing the child's anemia. Although dietary diversity plays a major role in anemia among children in sub-Saharan Africa, there is little evidence of an association between the dietary diversity and anemia level to identified potential strategies for prevention of anemia level in sub-Saharan Africa.

### Objective

To examine the association between dietary diversity and anemia levels among children aged 6–23 months in sub-Saharan Africa.

### Methods

The most recent Demographic and Health Surveys from 32 countries in SSA were considered for this study, which used pooled data from those surveys. In this study, a total weighted sample of 52,180 children aged 6–23 months was included. The diversity of the diet given to children was assessed using the minimum dietary diversity (MDD), which considers only four of the seven food groups. A multilevel ordinal logistic regression model was applied due to the DHS data's hierarchical structure and the ordinal nature of anemia. With

**Funding:** The authors received no specific funding for this work.

**Competing interests:** The authors have declared that no competing interests exist.

**Abbreviations:** AIC, Akaike Information Criteria; AMDD, Adequate Minimum Dietary Diversity; AOR, Adjusted Odds Ratio; BIC, Bayesian Information Criteria; CI, Confidence Interval; CRM, Continuous Ratio Model; DHS, Demographic health survey; DNA, Deoxyribonucleic Acid; EAs, Enumeration areas; HAZ, Z-score for Height-for-Age; ID, Iron Deficiency; IUGR, Intra uterine growth restriction; KR, Kids Record; LLR, Log likelihood ratio; LR, Likelihood ratio; MDD, Minimum Dietary Diversity; POM, Proportional Odds Model; PPOM, Partial Proportional Odds Model; PPOMWR, Partial Proportional Odds Model Without Restrictions; SD, Standard Deviation; SDG, Sustainable Development Goal; SM, Stereotype Model; SSA, Sub-Saharan Africa; WAZ, Z-score for Weight-for-Age; WHO, World Health Organizations; WHZ, Z-score for Weight-for-Height.

a p-value of 0.08, the Brant test found that the proportional odds assumption was satisfied. In addition, model comparisons were done using deviance. In the bi-variable analysis, variables having a p-value $\leq$0.2 were taken into account for multivariable analysis. The Adjusted Odds Ratio (AOR) with 95% Confidence Interval (CI) was presented for potential determinants of levels of anemia in the multivariable multilevel proportional odds model.

## Results

The overall prevalence of minimum dietary diversity and anemia among children aged 6–23 months were 43% [95% CI: 42.6%, 43.4%] and 72.0% [95% CI: 70.9%, 72.9%] respectively. Of which, 26.2% had mild anemia, 43.4% had moderate anemia, and 2.4% had severe anemia. MDD, being female child, being 18–23 months age, born from mothers aged $\geq$25, taking drugs for the intestinal parasite, higher level of maternal education, number of ANC visits, middle and richer household wealth status, distance of health facility and being born in Central and Southern Africa were significantly associated with the lower odds of levels of anemia. Contrarily, being 9–11- and 12–17-months age, size of child, having fever and diarrhea in the last two weeks, higher birth order, stunting, wasting, and underweight and being in West Africa were significantly associated with higher odds of levels of anemia.

## Conclusion

Anemia was a significant public health issue among children aged 6–23 months in sub-Saharan Africa. Minimum dietary diversity intake is associated with reduced anemia in children aged 6 to 23 months in sub-Saharan Africa. Children should be fed a variety of foods to improve their anemia status. Reducing anemia in children aged 6–23 months can be achieved by raising mother education levels, treating febrile illnesses, and improve the family's financial situation. Finally, iron fortification or vitamin supplementation could help to better reduce the risk of anemia and raise children's hemoglobin levels in order to treat anemia.

## Background

Micronutrient deficiency conditions are widespread worldwide specifically it is the leading cause of child mortality in sub-Saharan Africa [1,2]. Anemia is one of the micronutrient deficiency disorders that have global public health implications. It is prevalent in children and has short- and long-term effects on health and development, such as increasing the rate of growth retardation, decreasing immunity and intelligence, and even affecting an individual's health in adulthood [3,4]. It is a condition in which the number of red blood cells in the body is lower than the normal range, resulting in decreased oxygen availability to the tissues to meet physiological needs. The World Health Organization (WHO) defines anemia in children as having a hemoglobin level less than 11 g/dl [1]. Although anemia affects people of all ages, children under the age of two are the most vulnerable [5,6]. The child's rapid physical and mental development during the first two years of life demands the highest dietary requirements [7].

A total of 269 million children worldwide were expected to have anemia as of 2019 [8,9], with approximately two-thirds living in Asia and Africa. Sub-Saharan Africa was shown to have the highest anemia prevalence among children (46%–66%) [10–12].

Childhood anemia affects physical growth, mental development, and productivity in both the short- and long-term [13–15]. It results mental retardation [16], poor physical

performance [17], and poor motor development and control [18]. In the long term, it leads to reduced academic achievement [19–22]. Anemia can have dietary, medicinal, or hereditary causes [22]. According to findings from earlier research, a variety of factors can contribute to childhood anemia. These include: Infectious diseases (malaria, hookworm) [23], blood disorders (sickle cell anemia and thalassemia) [24], micronutrient deficiencies (folic acid, zinc, and vitamin B12) [25], iron deficiency [26], birth order [27], residence [28], child age [29–31], place of delivery [32,33], deworming [34], childhood nutritional status [35], household wealth status [36], maternal education [37], and maternal anemia [38,39]. Additionally, socioeconomic factors have an impact on anemia in emerging nations [9,40].

Despite significant improvements in the socioeconomic and health status of the population worldwide, sub-Saharan African nations continue to experience high rates of child mortality [41]. About 76.6% of children aged 6–23 months in sub-Saharan Africa are anemic and responsible for 5–18% of children mortality [1]. Producing sufficient evidence on the associated factors of individual, household, and community-level factors of anemia is necessary to develop timely interventions in anemia prevention and treatment, which is highly crucial for achieving the targets of the Sustainable Development Goals (SDGs)-2030 of reducing child mortality. Studies on the prevalence of anemia and its contributing factors in children in sub-Saharan Africa aged 6–23 months have been undertaken. However, because the effects of anemia vary based on the severity level of anemia (non-anemic, mild, moderate, and severe anemia), these studies are unable to capture the relationship between dietary diversity and ordinal nature of anemia status. As a result, we used multilevel ordinal logistic regression model to provide an accurate estimate and prevent information loss, as well as bi-variable analysis to examine the relationship between dietary diversity and anemia status.

This study is important for methodological and public health reasons. From the point of view of public health, this study's use of a relatively large sample size and pooled DHS data from 32 sub-Saharan African countries may have increased the study's power and allowed for generalization of the estimate. Additionally, the neighborhood effect can be considered when using a multilevel technique, and the outcome can provide an overall picture of SSA. From a methodological standpoint, it is clear that prior literature treated anemia as a binary outcome by categorizing it as either yes or no [3,42–45]. However, it is clear that treating mild, moderate, and severe anemia as yes is not statistically appropriate because there is a loss of information because the factor that can cause mild anemia may not be the same as the factor that can cause severe anemia. As a result, this study found the prevalence of dietary diversity and level of anemia in 32 sub-Saharan African children aged 6–23 months and explained the relationships between dietary diversity and level of anemia.

## Methods

### Data source, study setting and population

This study was based on the most recent Demographic and Health Survey (DHS) data of 32 SSA countries. DHS is a nationally representative survey routinely conducted every five years and collects data on basic health indicators like mortality, morbidity, fertility, and maternal and child health-related characteristics. The DHS adopted a cross-sectional design relying on a two-stage stratified sampling technique to recruit its respondents [46]. In the first stage, Enumeration Areas (EAs) were randomly selected based on the country's recent population and using the housing census as a sampling frame, households were randomly selected in the second stage. Each country's survey consists of different datasets including men, women, children, birth, and household datasets.

To assess the anemia status of children, hemoglobin testing was carried out among children in the selected households using HemoCue rapid testing methodology. For the test, a drop of capillary blood was taken from a child's fingertip or heel and was drawn into the micro cuvette which was then analyzed using the photometer that displays the hemoglobin concentration. There are different datasets in DHS and for this study; we used the Kids Record (KR) file. We extracted the data from the KR dataset based on literature and then appended using the STATA command "append using". A sample of 52,180 mother-child pairs of children within the ages of 6–23 months and mothers aged 15–49 years were included in this study (Table 1). Detailed information about DHS methodology can be found from the official database https://dhsprogram.com/Methodology/index.cfm.

## Ethical consideration

There was no need for ethical approval or participant consent because this study was based on publicly available survey data from the MEASURE DHS program. We have received permission from http://www.dhsprogram.com to download and utilize the data for this study. There are no recorded residential addresses or individual names in the datasets.

## Study variables and measurements

**Dependent variable.** The dependent variable of this study was the anemic status of children, which is an ordered categorical variable categorized into four ordinal categories; mildly anemic (hemoglobin level 10.0–10.9g/dl, moderately anemic (hemoglobin level 7.0–9.9g/dl), severely anemic (hemoglobin level <7.0g/dl), and not anemic (hemoglobin ≥11.0 g/dl). It was assessed based on the hemoglobin concentration in blood adjusted to the altitude. In DHS, before determining a child is anemic or not, they take into account altitude. Then, they have adjusted; the Hgb adjustment was made using the formula [47];

$$Hb_{adjust}(g/l) = -0.32 * (altitude\ in\ metres * 0.0033) + 0.22 * (altitude\ in\ meteres * 0.0033)^2$$

The adjustment for altitude was done to take into account the reduction in oxygen saturation of the blood.

**Independent variables.** The main independent variable was an indicator of dietary diversity called minimum dietary diversity (MDD). When children aged 6–23 months receive four or more of the seven dietary groups, it is said to have MDD [48–50]. Along with breast milk, the children were required to eat from at least four of the seven food categories. The seven food classes were: cereals, roots, and tubers; legumes and nuts; dairy products (milk, yogurt, cheese); flesh foods (meat, fish, fowl, liver, or other organs); eggs; fruits and vegetables high in vitamin A; and other fruits and vegetables. The scores for each food group, which ranged from 0 to 7, were added up to estimate MDD. A score of one (1) was given to any child who consumed any of the food groups, and a score of zero (0) was given to any child who did not. Children who consumed at least four (≥4) of the food groups were considered to have an adequate MDD, and this was coded as "1", whereas the remaining children who consumed fewer than four food groups were coded as "0 = inadequate". This classification and categorization was inspired by studies on either dietary diversity by itself or its association to under-nutrition [48–50].

Other explanatory variables were added as covariates as well. These factors were chosen based on their availability in the DHS dataset as well as their significant associations with children's anemic levels in the literature [1,12]. Given the hierarchical structure of the DHS data, where mothers and children were nested within the cluster, and the objectives of the study, two levels of independent variables were taken into consideration. Household-related

**Table 1. Description of the study samples for this study, and survey years.**

| Sub-Saharan Region | Country | Weighted N | Weighted % | Year of survey |
|---|---|---|---|---|
| **East Africa** | Burundi | 2049 | 3.9 | 2016–17 |
| | Ethiopia | 2878 | 5.5 | 2016 |
| | Madagascar | 1957 | 3.8 | 2008/09 |
| | Malawi | 1670 | 3.2 | 2015–16 |
| | Mozambique | 1720 | 3.3 | 2011 |
| | Rwanda | 1236 | 2.4 | 2014–15 |
| | Tanzania | 3395 | 6.5 | 2015–16 |
| | Uganda | 1560 | 3.0 | 2016 |
| | Zambia | 2948 | 5.6 | 2018 |
| | Zimbabwe | 1637 | 3.1 | 2015 |
| **Central Africa** | Angola | 2229 | 4.3 | 2015–16 |
| | DR. Congo | 2701 | 5.2 | 2013–14 |
| | Congo | 1655 | 3.2 | 2013 |
| | Cameroon | 1583 | 3.0 | 2018 |
| | Gabon | 1303 | 2.5 | 2012 |
| | SaoTome and principle | 567 | 1.1 | 2008/09 |
| **South Africa** | Lesotho | 493 | 0.9 | 2014 |
| | Namibia | 686 | 1.3 | 2013 |
| | Swaziland | 763 | 1.5 | 2005 |
| | South Africa | 293 | 0.6 | 2016 |
| **West Africa** | Burkina Faso | 2209 | 4.2 | 2010 |
| | Benin | 2141 | 4.1 | 2018 |
| | Cote d'Ivoire | 1053 | 2.0 | 2011–12 |
| | Ghana | 887 | 1.7 | 2014 |
| | Gambia | 1172 | 2.2 | 2019–20 |
| | Guinea | 1206 | 2.3 | 2018 |
| | Mali | 1404 | 2.7 | 2018 |
| | Nigeria | 3884 | 7.4 | 2018 |
| | Niger | 1543 | 3.0 | 2012 |
| | Sierra Leone | 1401 | 2.7 | 2019 |
| | Togo | 1093 | 2.1 | 2013–14 |
| | Liberia | 864 | 1.7 | 2019–20 |
| | **All countries** | **52,180** | **100** | |

characteristics, maternal-related, and child-related characteristics were the three categories of individual-level factors taken into consideration. Household-related factors were household wealth status, source of drinking water, sex of household head, and media exposure. Maternal related factors were maternal age, maternal education, marital status, maternal anemia, the number of Antenatal Care (ANC) visits during pregnancy, place of delivery, taking an iron supplement during pregnancy, wanted birth, mothers current employment status, and maternal smoking status. Among child-related factors; the age of a child, size of child at birth, sex of a child, birth order, diarrhea in the last two weeks, fever in the last two weeks, cough in the last two weeks, taking the drug for the intestinal parasite in the last six months, vitamin A supplementation in the last 6 months, wasting status (Z-scores for Weight-for-Height (WHZ)), underweight status (Z-scores for Weight-for-Age (WAZ)) and stunting status (Z-scores for

Height-for-Age (HAZ)). Region of sub-Saharan Africa, distance from health facility, and residence all fell under Level 2 (community-level variables).

The frequency of listening to the radio, watching television, and reading newspapers or magazines were the three variables used to measure media exposure. In this study, women who watched television, listened to radio, or read a newspaper or magazine at least once a week were categorized as having media exposure (coded "Yes") and those who did not were categorized as not having media exposure (coded "No"). Stunting is defined as children with a height-for-age Z-score (HAZ) $< -2SD$, and wasting as children with a weight-for-height Z-score (WHZ) $< -2SD$ and underweight children are those whose weight-for-age Z-score (WAZ) is $< -SD$ [51]. Mild, moderate, and severe maternal anemia were determined to be 10–11.9 g/dl, 7–9.9 g/dl, and 7 g/dl, respectively, for non-pregnant women and 10–10.9 g/dl, 7–9.9 g/dl, and 7 g/dl, respectively, for pregnant women.

## Data management and statistical analysis

STATA version 14.1 was used to pool, recode, and analyze data from 32 Sub-Saharan African countries. Prior to statistical analysis, the data were weighted using sampling weight, primary sampling unit, and strata to restore the survey's representativeness and provide accurate statistical estimates. Using frequencies and percentages, descriptive results were provided. The prevalence of MDD and anemia levels was summarized using forest plots. To assess the association between minimum dietary diversity and anemia among children, a bi-variable analysis was fitted, and the results was presented using odds ratio.

Due to the ordinal nature of the outcome variable (non-anemic, mild, moderate, and severe anemic), ordinal logistic regression model was applied. The ordinal logistic regression model that is most frequently used in epidemiological studies is the Proportional Odds Model (Cumulative Logit Model). The Proportional Odds (PO) assumptions, which state that the effects of all independent variables are constant across categories of the outcome variable, have been examined in order to select the best ordinal model for the data. The proportional odds assumption was then put to the test using the Brant test after fitting the proportional odds model. It tests the null hypothesis that the effects of independent variables are the same regardless of the anemia levels. The proportionate odds assumption was fulfilled, according to the Brant test (p = 0.08). So, to assessing the association between anemia and independent variables, we employed the proportional odds model.

Additionally, the DHS data is hierarchical in nature. Because mothers and children nested within a cluster, we might infer that study subjects within one cluster may have traits in common with those of subjects inside another cluster. This goes against the ordinal logistic regression model's independence observations and equal variance between clusters assumptions. This suggests that a sophisticated model must be used to account for the variation between clusters. In order to obtain a trustworthy estimate and standard error, a multilevel proportional odds model was used. This suggests the requirement for an advanced model that accounts for the heterogeneity between clusters. Consequently, a multilevel proportional odds model was performed to get a trustworthy estimate and standard error.

As a result, because the Brant test was satisfied, the multilevel proportional odds model provided a single Odds Ratio (OR) for an explanatory variable (severe vs moderate/mild/non-anemia, severe/moderate vs mild/non-anemia, and severe/moderate/mild vs non-anemic). To assess the variation in anemia among clusters, the Likelihood Ratio (LR) test, Intra-class Correlation Coefficient (ICC), and Median Odds Ratio (MOR) were computed. The ICC measures the degree of anemia heterogeneity between cluster (i.e., what proportion of anemia's overall

observed variation may be attributed to differences between clusters) [52].

$$ICC = \frac{\sigma^2}{\left(\sigma^2 + \frac{\pi^2}{3}\right)}$$

Where: the standard logit distribution has a variance of $\frac{\pi^2}{3}$, $\sigma_\mu^2$ indicates the cluster variance.

The MOR quantifies the variation or heterogeneity in anemia between clusters in terms of odds ratio scale and is defined as the median value of the odds ratio between the cluster at high likelihood of anemia and cluster at lower risk when randomly picking out individuals from two clusters (EAs) [53,54].

$$MOR = \exp\left(\sqrt{2\hat{\tau}^2} * \Phi^{-1}(0.75)\right)$$

Where $\hat{\tau}^2$ is the estimated variance of the distribution of the random effects, and $\Phi$ denotes the cumulative distribution function of the standard normal distribution, while $\Phi^{-1}$ (0.75) = 0.6745 is the 75[th] percentile of a standard normal distribution [54,55].

Four models were constructed for the multilevel logistic regression analysis. The first model, a null model without explanatory factors, was used to assess the degree of cluster variation in children anemia level; the second model was adjusted using individual-level variables; the third model was adjusted for community-level factors whereas the fourth was fitted with both individual and community level variables simultaneously. Since the models were nested, a model's deviance was compared using the -2Log-Likelihood Ratio (LLR), and the model with the lowest deviance was determined to be the best fit for the data.

For the multivariable multilevel proportional odds model, variables with a p-value ≤0.2 in the bi-variable model were taken into account. The Adjusted Odds Ratio (AOR) with 95% Confidence Interval (CI) were provided in the multivariable multilevel proportional odds model to indicate the strength of association, and the statistical significance for the final model was established at p-value ≤0.05.

## Results

### Descriptive characteristics of the study participants

A total of 52,180 children aged 6–23 months were included. Of these, 36,374 (69.7%) were from rural areas; among them were 21,050 (40.3%) from East Africa, 18,857 (36.1%) from West Africa, 2,235 (4.3%) from Southern Africa, and 10,038 (19.3%) from Central Africa. Around 26,416 (50.6%) males and 18,701 (35.8%) children between the ages of 12 and 17 were present. The majority (71.6%) of the children were delivered in a health facility, and 8,726 (16.7%) were small size at birth. Nearly half of the mothers of the children, 23,515 (45.1%), were between the ages of 25 and 34, and 17,954 (34.4%) were mothers who did not attain formal education. About 12,668 (24.3%), and 13,701 (26.3%) had diarrhea and fever in the last two weeks, respectively. In terms of nutritional status, the prevalence of stunted, underweight, and wasted children was 28%, 25%, and 9.4%, respectively (Table 2).

### Prevalence of minimum dietary diversity and levels of anemia

In SSA as a whole, there were 43% of children with adequate MDD, with Swaziland having the highest prevalence (55%) and Burkina Faso having the lowest prevalence (37%) (Fig 1). According to other findings, 26.9% of the children received grains, roots, or tubers. Sao Tome and Principe had the largest intake of grains, roots, and tubers (56.8%), while Burkina Faso had the lowest consumption (3.4%) (Table 3). The overall prevalence of anemia in children

**Table 2. Descriptive characteristics of children aged 6–23 months in sub-Saharan Africa.**

| Variables | Frequency (N = 52,180) | Percentage (%) |
|---|---|---|
| **Minimum dietary diversity** | | |
| Inadequate | 29,723 | 57.0 |
| Adequate | 22,457 | 43.0 |
| **Household Characteristics** | | |
| **Household wealth status** | | |
| Poor | 24,553 | 47.1 |
| Middle | 10,361 | 19.8 |
| Rich | 17,266 | 33.1 |
| **Source of drinking water supply** | | |
| Not improved | 22,907 | 43.9 |
| Improved | 29,273 | 56.1 |
| **Sex of household head** | | |
| Male | 41,065 | 78.7 |
| Female | 11,115 | 21.3 |
| **Media Exposure** | | |
| No | 19,076 | 36.6 |
| Yes | 33,104 | 63.4 |
| **Maternal related characteristics** | | |
| **Maternal age (in year)** | | |
| 15–24 | 18,458 | 35.4 |
| 25–34 | 23,515 | 45.1 |
| 35–49 | 10,207 | 19.5 |
| **Maternal education (52,178)** | | |
| No formal education | 17,954 | 34.4 |
| Primary | 18,665 | 35.8 |
| Secondary | 13,976 | 26.8 |
| Higher | 1,583 | 3.0 |
| **Marital status** | | |
| Never married | 4,488 | 8.6 |
| Married | 44,572 | 85.4 |
| Divorced/widowed/separated | 3,120 | 6.0 |
| **Maternal anemia level (n = 49,503)** | | |
| Severe | 420 | 0.9 |
| Moderate | 6,486 | 13.1 |
| Mild | 13,752 | 27.8 |
| No anemia | 28,845 | 58.2 |
| **Number of ANC visit during pregnancy** | | |
| No | 4,573 | 8.8 |
| **1–3** | 16,476 | 31.6 |
| $\geq 4$ | 31,131 | 59.6 |
| **Place of delivery** | | |
| Home | 15,428 | 29.6 |
| Health facility | 36,752 | 70.4 |
| **Taking iron supplements during pregnancy** | | |
| No | 10,336 | 19.8 |
| Yes | 41,844 | 80.2 |
| **Wanted birth** | | |

(*Continued*)

**Table 2.** (Continued)

| Variables | Frequency (N = 52,180) | Percentage (%) |
|---|---|---|
| No | 3,238 | 6.2 |
| Yes | 48,942 | 93.8 |
| **Mothers current employment status** | | |
| Not working | 15,446 | 29.6 |
| Working | 36,734 | 70.4 |
| **Mother's smoking cigarette** | | |
| No | 52,121 | 99.9 |
| Yes | 59 | 0.1 |
| **Child's characteristics** | | |
| **Sex of child** | | |
| Male | 26,416 | 50.6 |
| Female | 25,764 | 49.4 |
| **Age of child (in months)** | | |
| 6–8 | 8,964 | 17.2 |
| 9–11 | 8,813 | 16.9 |
| 12–17 | 18,701 | 35.8 |
| 18–23 | 15,702 | 30.1 |
| **Size of child at birth** | | |
| Large | 17,761 | 34.1 |
| Average | 25,693 | 49.2 |
| Small | 8,726 | 16.7 |
| **Birth order** | | |
| 1–3 | 30,216 | 57.9 |
| 4–6 | 15,623 | 29.9 |
| >6 | 6,341 | 12.2 |
| **Diarrhea in the last two weeks (n = 52,125)** | | |
| No | 39,457 | 75.7 |
| Yes | 12,668 | 24.3 |
| **Cough in the last two weeks (n = 52,136)** | | |
| No | 38,646 | 74.1 |
| Yes | 13,490 | 25.9 |
| **Fever in the last two weeks (n = 52,138)** | | |
| No | 38,437 | 73.7 |
| Yes | 13,701 | 26.3 |
| **Taking drug for intestinal parasite in the last 6 months (n = 49,542)** | | |
| No | 30,724 | 62.0 |
| Yes | 18,818 | 38.0 |
| **Stunting status** | | |
| Normal | 37,579 | 72.0 |
| Stunted | 14,601 | 28.0 |
| **Underweight status** | | |
| Normal | 39,115 | 75.0 |
| Underweight | 13,065 | 25.0 |
| **Wasting status** | | |
| Normal | 47,261 | 90.6 |
| Wasted | 4,919 | 9.4 |
| **Community-level characteristics** | | |

(*Continued*)

**Table 2.** (Continued)

| Variables | Frequency (N = 52,180) | Percentage (%) |
|---|---|---|
| **Distance of health facilities (n = 52,162)** | | |
| Big problem | 21,882 | 41.9 |
| Not a big problem | 30,281 | 58.1 |
| **Residence** | | |
| Rural | 36,374 | 69.7 |
| Urban | 15,806 | 30.3 |
| **Region** | | |
| East Africa | 21,050 | 40.3 |
| West Africa | 18,857 | 36.1 |
| Southern Africa | 2,235 | 4.3 |
| Central Africa | 10,038 | 19.3 |

aged 6–23 months was 72.0% [95% CI: 70.9%, 72.9%]. In this study, 26.2% [95% CI: 25.8%, 26.6%] of children aged 6–23 months had mild anemia (Fig 2), 43.4% [95% CI: 42.9%, 43.8.2%] moderate anemia (Fig 3) and 2.4% [95% CI: 2.2%, 2.5%] severe anemia (Fig 4).

As seen in Fig 2, the highest prevalence of mild anemia was recorded by Gabon (34.8%), while the lowest prevalence was found in Burkina Faso (12.0%). As opposed to this, Burkina Faso had the highest prevalence of moderate anemia (64.2%) and severe anemia (16.2%), whereas Rwanda had the lowest prevalence (23.4% and 0.5%, respectively) (Figs 3 and 4).

## Association between minimum dietary diversity and anemia among children in SSA

Table 4 shows the results of the association between minimum dietary diversity (MDD) and anemia among children. Based on this analysis using dietary diversity score and MDD as independent variables, diverse dietary intake was significantly associated with reduced risk of anemia in children. It has been shown that the more food groups they diet, the less likely they are to develop anemia. Therefore, both dietary diversity scores and MDD analyzes showed that children who ate a more varied diet were less likely to develop anemia than those who ate a less varied diet. In the model, children who received MDD were significantly less likely to be anemic [OR = 0.95, 95% CI; 0.92, 0.97] compared with children who did not receive MDD. As a result, we should further investigate the relationship between the minimum dietary diversity of children and their level of anemia in children aged 6–23 months.

## Model fit statistics

As a result, the odds ratio results from the models used to compare anemic to non-anemic and severely anemic to mildly/moderately/non-anemic had the same interpretations. In the null model, the ICC value was 3.52% [95% CI: 2.95%, 4.09%], indicating that 3.52% of the total variability of anemia level was attributable to differences between clusters, while the remaining 96.48% of the total variability of anemia level was due to individual differences. Additionally, the MOR in the null model was 1.38 (95% CI: 1.35, 1.42). The final model was the one with the best fit for the data, as indicated by its lowest deviance value (Table 5).

| Country | AMDD | Total | Prevalence(95% CI) | % Weight |
|---|---|---|---|---|
| Angola | 1011 | 2229 | 45.36 (43.29, 47.42) | 4.20 |
| Burkina Faso | 824 | 2209 | 37.30 (35.29, 39.32) | 4.41 |
| Benin | 908 | 2141 | 42.41 (40.32, 44.50) | 4.10 |
| Burundi | 825 | 2049 | 40.26 (38.14, 42.39) | 3.98 |
| Dr Congo | 1097 | 2701 | 40.61 (38.76, 42.47) | 5.23 |
| Congo | 679 | 1655 | 41.03 (38.66, 43.40) | 3.20 |
| Cote de vaire | 417 | 1053 | 39.60 (36.65, 42.56) | 2.06 |
| Cameroon | 744 | 1583 | 47.00 (44.54, 49.46) | 2.97 |
| Ethiopia | 1188 | 2878 | 41.28 (39.48, 43.08) | 5.55 |
| Gabon | 564 | 1303 | 43.28 (40.59, 45.97) | 2.48 |
| Ghana | 352 | 887 | 39.68 (36.46, 42.90) | 1.73 |
| Gambia | 475 | 1172 | 40.53 (37.72, 43.34) | 2.27 |
| Guinea | 546 | 1206 | 45.27 (42.46, 48.08) | 2.27 |
| Liberia | 368 | 864 | 42.59 (39.30, 45.89) | 1.65 |
| Lesotho | 187 | 493 | 37.93 (33.65, 42.21) | 0.98 |
| Madagascar | 897 | 1957 | 45.84 (43.63, 48.04) | 3.68 |
| Mali | 601 | 1404 | 42.81 (40.22, 45.39) | 2.68 |
| Malawi | 743 | 1670 | 44.49 (42.11, 46.87) | 3.16 |
| Mozambique | 776 | 1720 | 45.12 (42.76, 47.47) | 3.25 |
| Nigeria | 1859 | 3884 | 47.86 (46.29, 49.43) | 7.27 |
| Niger | 597 | 1543 | 38.69 (36.26, 41.12) | 3.04 |
| Namibia | 342 | 686 | 49.85 (46.11, 53.60) | 1.28 |
| Rwanda | 498 | 1236 | 40.29 (37.56, 43.03) | 2.40 |
| Sierra Leone | 644 | 1401 | 45.97 (43.36, 48.58) | 2.64 |
| Saotome and Principe | 269 | 567 | 47.44 (43.33, 51.55) | 1.06 |
| Swaziland | 438 | 763 | 57.40 (53.90, 60.91) | 1.46 |
| Togo | 457 | 1093 | 41.81 (38.89, 44.74) | 2.10 |
| Tanzania | 1326 | 3395 | 39.06 (37.42, 40.70) | 6.66 |
| Uganda | 671 | 1560 | 43.01 (40.56, 45.47) | 2.97 |
| South Africa | 162 | 293 | 55.29 (49.60, 60.98) | 0.55 |
| Zambia | 1260 | 2948 | 42.74 (40.96, 44.53) | 5.63 |
| Zimbabwe | 732 | 1637 | 44.72 (42.31, 47.12) | 3.09 |
| Pooled pervalence of AMDD | | | 42.98 (42.56, 43.41) | 100.00 |

0   50

**Fig 1. The forest plot showing the prevalence of adequate minimum dietary diversity in sub-Saharan Africa.**

**Table 3. Distribution of the types of food groups consumed by the children per country.**

| Countries | Grains, roots And tubers | Legumes and nuts | Dairy products | Flesh foods | Eggs | Vitamin A- rich fruits and vegetables | Other fruits and vegetables |
|---|---|---|---|---|---|---|---|
| Angola | 28.8 | 15.2 | 6.8 | 8.3 | 13.0 | 25.3 | 27.5 |
| Burkina Faso | 3.4 | 15.3 | 0.5 | 9.8 | 4.8 | 3.1 | 5.0 |
| Benin | 30.3 | 14.2 | 15.4 | 14.8 | 20.3 | 12.9 | 26.8 |
| Burundi | 40.6 | 13.4 | 1.0 | 7.9 | 5.1 | 43.7 | 7.9 |
| Dr Congo | 29.9 | 16.1 | 0.6 | 5.2 | 8.1 | 15.4 | 23.5 |
| Congo | 51.5 | 14.2 | 3.3 | 26.7 | 6.0 | 19.5 | 14.6 |
| Cote de vaire | 38.3 | 13.8 | 1.2 | 7.8 | 8.4 | 9.7 | 10.3 |
| Cameroon | 20.3 | 14.9 | 2.8 | 9.6 | 13.7 | 20.1 | 36.7 |
| Ethiopia | 17.6 | 17.2 | 14.3 | 28.4 | 15.0 | 12.2 | 9.7 |
| Gabon | 37.5 | 13.3 | 5.9 | 29.2 | 12.9 | 6.6 | 12.8 |
| Ghana | 29.3 | 15.8 | 1.1 | 11.7 | 20.8 | 5.3 | 18.1 |
| Gambia | 11.5 | 13.6 | 2.8 | 27.6 | 10.4 | 1.5 | 16.6 |
| Guinea | 13.1 | 11.6 | 7.6 | 19.4 | 23.0 | 29.0 | 8.9 |
| Liberia | 24.5 | 13.4 | 3.4 | 5.2 | 5.7 | 6.5 | 11.1 |
| Lesotho | 11.2 | 12.7 | 1.8 | 18.7 | 29.9 | 0.6 | 16.7 |
| Madagascar | 38.6 | 18.6 | 4.6 | 10.9 | 6.9 | 10.7 | 37.1 |
| Mali | 11.5 | 13.4 | 4.5 | 27.8 | 13.7 | 5.7 | 12.1 |
| Malawi | 13.9 | 19.9 | 1.8 | 4.9 | 13.1 | 45.9 | 28.8 |
| Mozambique | 37.3 | 14.7 | 6.0 | 4.7 | 15.7 | 15.2 | 32.0 |
| Nigeria | 32.0 | 19.8 | 11.3 | 15.2 | 17.2 | 4.8 | 15.6 |
| Niger | 15.5 | 16.7 | 9.4 | 11.2 | 6.5 | 16.4 | 7.3 |
| Namibia | 26.5 | 16.2 | 18.8 | 13.5 | 18.4 | 13.5 | 30.6 |
| Rwanda | 44.2 | 16.7 | 0.9 | 27.3 | 6.4 | 45.2 | 24.0 |
| Sierra Leone | 30.1 | 15.2 | 15.7 | 22.2 | 19.8 | 27.5 | 24.8 |
| Sao tome and principle | 56.8 | 12.9 | 17.1 | 25.8 | 22.8 | 33.3 | 34.3 |
| Swaziland | 37.0 | 15.4 | 39.8 | 36.6 | 23.7 | 32.5 | 28.4 |
| Togo | 38.0 | 14.9 | 6.1 | 5.0 | 11.2 | 20.4 | 10.5 |
| Tanzania | 25.7 | 16.4 | 4.2 | 18.9 | 6.3 | 27.6 | 20.3 |
| Uganda | 38.1 | 17.2 | 1.7 | 24.6 | 14.4 | 14.4 | 20.5 |
| South Africa | 41.3 | 21.2 | 17.5 | 27.6 | 37.2 | 11.9 | 41.6 |
| Zambia | 11.5 | 16.1 | 1.3 | 7.1 | 19.6 | 12.4 | 26.6 |
| Zimbabwe | 16.6 | 13.9 | 2.4 | 6.7 | 14.0 | 13.2 | 25.9 |
| **All countries** | **26.9** | **15.8** | **6.3** | **15.1** | **13.0** | **17.5** | **20.1** |

## Multivariable multilevel proportional odds model analysis results

The bi-variable analysis was carried out to determine the factors that contribute to anemia. In accordance with this, minimum dietary diversity (MDD), sex of the child, age of the child, size of child at birth, birth order, diarrhea, fever, taking drugs for an intestinal parasite, stunting, underweight, wasting, maternal age, maternal education, number of ANC visits, household wealth status, place of delivery, media exposure, distance to the health facility, residence and region were taken into account for the multivariable analysis (p < 0.2). In the multivariable multilevel proportional odds model; receiving a minimum dietary diversity (MDD) adequately, sex of child, child aged (18–23 months), maternal age, taking drugs for the intestinal parasite in the last six months, maternal education, number of ANC visits, household wealth status, distance of health facility and being born in Central Africa and Southern Africa were

| Country | Mild Anemia | Total | Prevalence(95%CI) | % Weight |
|---|---|---|---|---|
| Angola | 743 | 2229 | 33.33 (31.38, 35.29) | 3.66 |
| Burkina Faso | 265 | 2209 | 12.00 (10.64, 13.35) | 7.63 |
| Benin | 551 | 2141 | 25.74 (23.88, 27.59) | 4.08 |
| Burundi | 552 | 2049 | 26.94 (25.02, 28.86) | 3.80 |
| Dr Congo | 652 | 2701 | 24.14 (22.53, 25.75) | 5.38 |
| Congo | 489 | 1655 | 29.55 (27.35, 31.74) | 2.90 |
| Cote de vaire | 239 | 1053 | 22.70 (20.17, 25.23) | 2.19 |
| Cameroon | 442 | 1583 | 27.92 (25.71, 30.13) | 2.87 |
| Ethiopia | 685 | 2878 | 23.80 (22.25, 25.36) | 5.79 |
| Gabon | 454 | 1303 | 34.84 (32.26, 37.43) | 2.09 |
| Ghana | 256 | 887 | 28.86 (25.88, 31.84) | 1.58 |
| Gambia | 287 | 1172 | 24.49 (22.03, 26.95) | 2.31 |
| Guinea | 410 | 1206 | 34.00 (31.32, 36.67) | 1.96 |
| Liberia | 228 | 864 | 26.39 (23.45, 29.33) | 1.62 |
| Lesotho | 117 | 493 | 23.73 (19.98, 27.49) | 0.99 |
| Madagascar | 665 | 1957 | 33.98 (31.88, 36.08) | 3.18 |
| Mali | 254 | 1404 | 18.09 (16.08, 20.10) | 3.45 |
| Malawi | 453 | 1670 | 27.13 (24.99, 29.26) | 3.08 |
| Mozambique | 447 | 1720 | 25.99 (23.92, 28.06) | 3.26 |
| Nigeria | 1087 | 3884 | 27.99 (26.57, 29.40) | 7.03 |
| Niger | 324 | 1543 | 21.00 (18.97, 23.03) | 3.39 |
| Namibia | 200 | 686 | 29.15 (25.75, 32.56) | 1.21 |
| Rwanda | 374 | 1236 | 30.26 (27.70, 32.82) | 2.14 |
| Sierra Leone | 441 | 1401 | 31.48 (29.05, 33.91) | 2.37 |
| Saotome and Principe | 179 | 567 | 31.57 (27.74, 35.40) | 0.96 |
| Swaziland | 197 | 763 | 25.82 (22.71, 28.92) | 1.45 |
| Togo | 262 | 1093 | 23.97 (21.44, 26.50) | 2.19 |
| Tanzania | 1012 | 3395 | 29.81 (28.27, 31.35) | 5.92 |
| Uganda | 415 | 1560 | 26.60 (24.41, 28.80) | 2.91 |
| South Africa | 76 | 293 | 25.94 (20.92, 30.96) | 0.56 |
| Zambia | 919 | 2948 | 31.17 (29.50, 32.85) | 5.01 |
| Zimbabwe | 432 | 1637 | 26.39 (24.25, 28.52) | 3.07 |
| Pooled prevalence of mild anmeia | | | 26.18 (25.81, 26.56) | 100.00 |

-50  0  50

**Fig 2. Forest plot showing the prevalence of mild anemia in sub-Saharan Africa.**

| Country | Moderate anemia | Total | Prevalence (95% CI) | % Weight |
|---|---|---|---|---|
| Angola | 881 | 2229 | 39.52 (37.49, 41.55) | 4.23 |
| Burkina Faso | 1418 | 2209 | 64.19 (62.19, 66.19) | 4.35 |
| Benin | 1074 | 2141 | 50.16 (48.05, 52.28) | 3.88 |
| Burundi | 800 | 2049 | 39.04 (36.93, 41.16) | 3.90 |
| Dr Congo | 1039 | 2701 | 38.47 (36.63, 40.30) | 5.17 |
| Congo | 738 | 1655 | 44.59 (42.20, 46.99) | 3.03 |
| Cote de vaire | 597 | 1053 | 56.70 (53.70, 59.69) | 1.94 |
| Cameroon | 681 | 1583 | 43.02 (40.58, 45.46) | 2.93 |
| Ethiopia | 1336 | 2878 | 46.42 (44.60, 48.24) | 5.24 |
| Gabon | 521 | 1303 | 39.98 (37.32, 42.64) | 2.46 |
| Ghana | 413 | 887 | 46.56 (43.28, 49.84) | 1.62 |
| Gambia | 431 | 1172 | 36.77 (34.01, 39.54) | 2.28 |
| Guinea | 536 | 1206 | 44.44 (41.64, 47.25) | 2.21 |
| Liberia | 422 | 864 | 48.84 (45.51, 52.18) | 1.57 |
| Lesotho | 143 | 493 | 29.01 (25.00, 33.01) | 1.08 |
| Madagascar | 605 | 1957 | 30.91 (28.87, 32.96) | 4.15 |
| Mali | 872 | 1404 | 62.11 (59.57, 64.65) | 2.70 |
| Malawi | 838 | 1670 | 50.18 (47.78, 52.58) | 3.03 |
| Mozambique | 748 | 1720 | 43.49 (41.15, 45.83) | 3.17 |
| Nigeria | 1809 | 3884 | 46.58 (45.01, 48.14) | 7.07 |
| Niger | 950 | 1543 | 61.57 (59.14, 64.00) | 2.95 |
| Namibia | 266 | 686 | 38.78 (35.13, 42.42) | 1.31 |
| Rwanda | 289 | 1236 | 23.38 (21.02, 25.74) | 3.13 |
| Sierra Leone | 617 | 1401 | 44.04 (41.44, 46.64) | 2.58 |
| Saotome and Principe | 252 | 567 | 44.44 (40.35, 48.53) | 1.04 |
| Swaziland | 254 | 763 | 33.29 (29.95, 36.63) | 1.56 |
| Togo | 627 | 1093 | 57.37 (54.43, 60.30) | 2.02 |
| Tanzania | 1388 | 3395 | 40.88 (39.23, 42.54) | 6.36 |
| Uganda | 641 | 1560 | 41.09 (38.65, 43.53) | 2.92 |
| South Africa | 110 | 293 | 37.54 (32.00, 43.09) | 0.57 |
| Zambia | 1162 | 2948 | 39.42 (37.65, 41.18) | 5.59 |
| Zimbabwe | 411 | 1637 | 25.11 (23.01, 27.21) | 3.94 |
| Pooled prevalence of moderate anemia | | | 43.41 (42.99, 43.82) | 100.00 |

**Fig 3. Forest plot showing the prevalence of moderate anemia in sub-Saharan Africa.**

| Country | Severe anemia | Total | Prevalence(95%CI) | % Weight |
|---|---|---|---|---|
| Angola | 85 | 2229 | 3.81 (3.02, 4.61) | 2.63 |
| Burkina Faso | 358 | 2209 | 16.21 (14.67, 17.74) | 0.70 |
| Benin | 114 | 2141 | 5.32 (4.37, 6.28) | 1.84 |
| Burundi | 78 | 2049 | 3.81 (2.98, 4.64) | 2.42 |
| Dr Congo | 123 | 2701 | 4.55 (3.77, 5.34) | 2.69 |
| Congo | 25 | 1655 | 1.51 (0.92, 2.10) | 4.81 |
| Cote de vaire | 63 | 1053 | 5.98 (4.55, 7.42) | 0.81 |
| Cameroon | 34 | 1583 | 2.15 (1.43, 2.86) | 3.26 |
| Ethiopia | 117 | 2878 | 4.07 (3.34, 4.79) | 3.19 |
| Gabon | 22 | 1303 | 1.69 (0.99, 2.39) | 3.40 |
| Ghana | 39 | 887 | 4.40 (3.05, 5.75) | 0.91 |
| Gambia | 29 | 1172 | 2.47 (1.59, 3.36) | 2.10 |
| Guinea | 73 | 1206 | 6.05 (4.71, 7.40) | 0.92 |
| Liberia | 29 | 864 | 3.36 (2.16, 4.56) | 1.15 |
| Lesotho | 15 | 493 | 3.04 (1.53, 4.56) | 0.72 |
| Madagascar | 11 | 1957 | 0.56 (0.23, 0.89) | 15.15 |
| Mali | 85 | 1404 | 6.05 (4.81, 7.30) | 1.07 |
| Malawi | 50 | 1670 | 2.99 (2.18, 3.81) | 2.49 |
| Mozambique | 92 | 1720 | 5.35 (4.29, 6.41) | 1.47 |
| Nigeria | 148 | 3884 | 3.81 (3.21, 4.41) | 4.58 |
| Niger | 59 | 1543 | 3.82 (2.87, 4.78) | 1.82 |
| Namibia | 9 | 686 | 1.31 (0.46, 2.16) | 2.29 |
| Rwanda | 6 | 1236 | 0.49 (0.10, 0.87) | 11.07 |
| Sierra Leone | 50 | 1401 | 3.57 (2.60, 4.54) | 1.76 |
| Saotome and Principe | 11 | 567 | 1.94 (0.80, 3.08) | 1.29 |
| Swaziland | 16 | 763 | 2.10 (1.08, 3.11) | 1.61 |
| Togo | 43 | 1093 | 3.93 (2.78, 5.09) | 1.25 |
| Tanzania | 77 | 3395 | 2.27 (1.77, 2.77) | 6.63 |
| Uganda | 49 | 1560 | 3.14 (2.28, 4.01) | 2.22 |
| South Africa | 12 | 293 | 4.10 (1.83, 6.36) | 0.32 |
| Zambia | 87 | 2948 | 2.95 (2.34, 3.56) | 4.45 |
| Zimbabwe | 13 | 1637 | 0.79 (0.36, 1.22) | 8.99 |
| Pooled prevalence of severe anemia | | | 2.35 (2.22, 2.48) | 100.00 |

-20    0    20

**Fig 4. Forest plot showing the prevalence of severe anemia in sub-Saharan Africa.**

**Table 4. Bivariable analysis of minimum dietary diversity and anemia among children aged 6–23 months in sub-Saharan Africa.**

| Variable | Categories | n (%) | OR [95% CI] |
|---|---|---|---|
| Dietary Score | 0 | 6,389(12.24) | Ref. |
| | 1 | 12,267(23.51) | 0.99 [0.93, 1.05] |
| | 2 | 7,776(14.90) | 0.84 [0.79, 0.89]* |
| | 3 | 3,291(6.31) | 0.73 [0.68, 0.79]* |
| | 4 | 1,435(2.75) | 0.77 [0.60, 0.74]* |
| | 5 | 11,583(22.20) | 0.97 [0.90, 1.01] |
| | 6 | 7,256(13.91) | 0.81 [0.77, 0.86]* |
| | 7 | 3,618(6.93) | 0.71 [0.64, 0.75]* |
| Minimum Dietary Diversity | Inadequate | 29,723(57%) | Ref. |
| | Adequate | 22,457(43%) | 0.95 [0.92, 0.97]* |

n: Sample weight, CI: Confidence interval, OR: Odds ratio.

significantly associated with the lower odds of severity levels of anemia. Contrarily, child age (9–11 and 12–17 months), size of child at birth, fever in the last two weeks, diarrhea in the last two weeks, birth order, stunting, wasting, and underweight, residence and being in West Africa were significantly associated with higher odds of severity levels of anemia.

Children with adequate dietary diversity were 6% less likely to have higher levels of anemia (AOR = 0.94, 95% CI: 0.92, 0.98), compared children with inadequate MDD. The odds of having higher levels of anemia among female children decreased by 23% (AOR = 0.77, 95% CI: 0.75, 0.79) compared to male children. The odds of having higher levels of anemia were decreased by 25% (AOR = 0.75, 95% CI: 0.71, 0.79) in children aged 18–23 months compared those aged 6–8 months. The odds of a higher level of anemia for children aged 9–11 and 12–17 months were 1.10 times [AOR = 1.10, 95% CI: 1.04, 1.16], 1.05 times [AOR = 1.05, 95% CI: 1.00, 1.11] higher than children aged 6–8 months, respectively. Children who were smaller size at birth and average size at birth were 1.25 times [AOR = 1.25, 95% CI: 1.19, 1.32], 1.11 times [AOR = 1.11, 95% CI: 1.07, 1.15] higher odds of higher levels of anemia than children larger size at birth, respectively.

Children being 4–6 birth order and >6 birth order had 1.10 times [AOR = 1.10, 95% CI: 1.05, 1.15], 1.17 times [AOR = 1.17, 95% CI: 1.09, 1.26] higher odds of higher levels of anemia compared to 1–3 birth order, respectively. Children who had diarrhea and fever in the last two weeks had 1.05 times [AOR = 1.05, 95% CI: 1.01, 1.09], and 1.46 times [AOR = 1.46, 95% CI: 1.40, 1.52] higher odds of a higher level of anemia compared to children who did not have diarrhea and fever, respectively. The odds of being at higher anemia status among children who took drugs for the intestinal parasite in the last six months were decreased by 10% [AOR = 0.90, 95% CI: 0.86, 0.93] than those who did not take drugs. Stunted, wasted and underweight children had 1.13 times [AOR = 1.13, 95% CI: 1.08, 1.18], 1.18 times [AOR = 1.18, 95% CI: 1.10, 1.26], and 1.16 times [AOR = 1.16, 95% CI: 1.10, 1.22] higher odds of higher level of anemia, respectively.

The odds of having higher level of anemia among children whose mother aged 25–34 years and 35–49 years were decreased by 17% [AOR = 0.83, 95% CI: 0.79, 0.87] and 28% [AOR = 0.72, 95% CI: 0.67, 0.77] compared to children whose mother aged 15–24 years, respectively. The odds of being at higher level of anemia among children whose mother education level at primary, secondary and higher were decreased by 19% [AOR = 0.81, 95% CI: 0.78, 0.85], 29% [AOR = 0.71, 95% CI: 0.68, 0.75], and 40% [AOR = 0.60, 95% CI: 0.54, 0.67]

**Table 5. Fixed and random effects result on the association between dietary diversity and anemia level among children aged 6–23 months in sub-Saharan Africa.**

| Variable | Null model | Model 1 | Model 2 | Model 3 |
|---|---|---|---|---|
| | | AOR with 95% CI | AOR with 95% CI | AOR with 95% CI |
| **Fixed effect model** | | | | |
| **Minimum dietary diversity** | | | | |
| Inadequate | | 1 | | 1 |
| Adequate | | 0.97 [0.96, 0.99] | | 0.94 [0.92, 0.98]* |
| **Child characteristics** | | | | |
| **Sex of child** | | | | |
| Male | | 1 | | 1 |
| Female | | 0.78 [0.75, 0.80] | | 0.77 [0.75, 0.79]* |
| **Age of child (months)** | | | | |
| 6–8 | | 1 | | 1 |
| 9–11 | | 1.08 [1.02, 1.15] | | 1.10 [1.04, 1.16]* |
| 12–17 | | 1.05 [1.01, 1.11] | | 1.05 [1.00, 1.11]* |
| 18–23 | | 0.75 [0.71, 0.79] | | 0.75 [0.71, 0.79]* |
| **Size of child at birth** | | | | |
| Large | | 1 | | 1 |
| Average | | 1.07 [1.03, 1.11] | | 1.11 [1.07, 1.15]* |
| Smaller | | 1.21 [1.15, 1.27] | | 1.25 [1.19, 1.32]** |
| **Birth order** | | | | |
| 1–3 | | 1 | | 1 |
| 4–6 | | 1.11 [1.06, 1.16] | | 1.10 [1.05, 1.15]* |
| >6 | | 1.16 [1.08, 1.25] | | 1.17 [1.09, 1.26]* |
| **Diarrhea in the last two weeks (n=52,125)** | | | | |
| No | | 1 | | 1 |
| Yes | | 1.02 [0.98, 1.06] | | 1.05 [1.01, 1.09]* |
| **Fever in the last two weeks (n=52,138)** | | | | |
| No | | 1 | | 1 |
| Yes | | 1.48 [1.42, 1.55] | | 1.46 [1.40, 1.52]** |
| **Taking drug for intestinal parasite in the last 6 months (n=49,542)** | | | | |
| No | | 1 | | 1 |
| Yes | | 0.88 [0.85, 0.92] | | 0.90 [0.86, 0.93]* |
| **Stunting status** | | | | |
| Normal | | 1 | | 1 |
| Stunted | | 1.06 [1.02, 1.11] | | 1.13 [1.08, 118]* |
| **Underweight status** | | | | |
| Normal | | 1 | | 1 |
| Underweight | | 1.22 [1.16, 1.29] | | 1.16 [1.10, 1.22]** |
| **Wasting status** | | | | |
| Normal | | 1 | | 1 |
| Wasted | | 1.21 [1.14, 1.30] | | 1.18 [1.10, 1.26]** |
| **Maternal characteristics** | | | | |
| **Maternal age (in year)** | | | | |
| 15–24 | | 1 | | 1 |
| 25–34 | | 0.81 [0.78, 0.85] | | 0.83 [0.79, 0.87]* |
| 35–49 | | 0.70 [0.65, 0.74] | | 0.72 [0.67, 0.77]* |
| **Maternal education (52,178)** | | | | |
| No formal education | | 1 | | 1 |

*(Continued)*

**Table 5.** (Continued)

| Variable | Null model | Model 1 | Model 2 | Model 3 |
|---|---|---|---|---|
| | | AOR with 95% CI | AOR with 95% CI | AOR with 95% CI |
| Primary | | 0.63 [0.60, 0.66] | | 0.81 [0.78, 0.85]* |
| Secondary | | 0.58 [0.55, 0.61] | | 0.71 [0.68, 0.75]* |
| Higher | | 0.48 [0.43, 0.53] | | 0.60 [0.54, 0.67]* |
| **Number of ANC visit during pregnancy** | | | | |
| No | | 1 | | 1 |
| 1–3 | | 1.04 [0.97, 1.12] | | 1.06 [0.99, 1.14] |
| ≥4 | | 0.89 [0.83, 0.95] | | 0.88 [0.82, 0.95]* |
| **Place of delivery** | | | | |
| Home | | 1 | | 1 |
| Health facility | | 1.00 [0.97, 1.05] | | 1.02 [0.98, 1.07] |
| **Household Characteristics** | | | | |
| **Household wealth status** | | | | |
| Poor | | 1 | | 1 |
| Middle | | 0.96 [0.92, 1.01] | | 0.95 [0.92, 0.99]* |
| Rich | | 0.89 [0.85, 0.93] | | 0.90 [0.85, 0.94]* |
| **Media Exposure** | | | | |
| No | | 1 | | 1 |
| Yes | | 1.08 [1.04, 1.13] | | 1.04 [0.99, 1.08] |
| **Community-level characteristics** | | | | |
| **Distance of health facilities (n=52,162)** | | | | |
| Big problem | | | 1 | 1 |
| Not a big problem | | | 0.91 [0.88, 0.94] | 0.97 [0.93, 0.99]* |
| **Residence** | | | | |
| Urban | | | 1 | 1 |
| Rural | | | 1.30 [1.25, 1.35] | 1.10 [1.05, 1.15]* |
| **Region** | | | | |
| Central Africa | | | 1 | 1 |
| East Africa | | | 0.85 [0.81, 0.89] | 0.83 [0.79, 0.88]* |
| West Africa | | | 1.72 [1.64, 1.81] | 1.57 [1.49, 1.66]** |
| Southern Africa | | | 0.69 [0.63, 0.75] | 0.69 [0.63, 0.76]* |
| **/cut1** | 0.32 [0.31, 0.33] | -1.71 [-1.81, -1.60] | -0.90 [-0.96, -0.85] | -1.51 [-1.64, -1.39] |
| **/cut2** | 0.08 [0.05, 0.11] | -0.46 [-0.56, -0.35] | 0.34 [0.28, 0.39] | -0.25 [-0.37, -0.12] |
| **/cut3** | 3.25 [3.20, 3.30] | 2.82 [2.71, 2.94] | 3.56 [3.49, 3.63] | 3.07 [2.94, 3.20] |
| **Random effect analysis result** | | | | |
| Community level variance | 0.12 [0.10, 0.14] | | | |
| LR-test | Prob >= chibar2 <0.001 | | | |
| ICC | 3.52% [2.95%, 4.09%] | | | |
| MOR | 1.38 [1.35, 1.42] | | | |
| LLR | -61,568.29 | -56,916.42 | -60,689.31 | -56,431.27 |
| Deviance (-2LLR) | 123,136.58 | 113,832.84 | 121,378.62 | 112,862.54 |

*p–value < 0.05,

**p–value < 0.01:

AOR = Adjusted Odds Ratio (proportionate odds ratio): CI: Confidence Interval: ICC = Intra-class Correlation Coefficient: LR = Likelihood Ratio: LLR = Log-likelihood Ratio: MOR: Median Odds Ratio.

compared to children whose mother had no formal education, respectively. Children born to mothers who attended four or more ANC visits experienced a 12% reduction in the likelihood of having greater levels of anemia [AOR = 0.88, 95% CI: 0.82, 0.95] than children born to mothers who had no ANC visits during pregnancy. The odds of having higher levels of anemia among children from middle and richer household wealth were decreased by 5% [AOR = 0.95, 95% CI: 0.92, 0.99] and 10% [AOR = 0.90, 95% CI: 0.85, 0.94] compared to children from poor household wealth, respectively.

The odds of having higher levels of anemia among children born to mothers who reported distance to a health facility as not a big problem were decreased by 3% [AOR = 0.97, 95% CI: 0.93, 0.99] compared to children who reported distance to a health facility as big problem. Children who born in rural area had 1.10 times [AOR = 1.10, 95% CI: 1.05, 1.15] higher odds of a higher level of anemia compared to children who born in urban area. The odds of being at higher level of anemia among children in East Africa and Southern Africa were decreased by 17% [AOR = 0.83, 95% CI: 0.79, 0.88] and 31% [AOR = 0.69, 95% CI: 0.63, 0.76] compared to children in Central Africa, respectively whereas children in West Africa had 1.57 times (AOR = 1.57, 95% CI: 1.49, 1.66) higher odds of higher levels of anemia than those in Central Africa (Table 5).

## Discussion

This study examined the association between minimum dietary diversity (MDD) and anemia level among children aged 6–23 months. The odds ratios for the final model appeared to have maintained constant across all cut-off points for the presence of childhood anemia, according to the Brant test of parallel odds assumption (p-value = 0.08 at a 5% level of significance). The association between other child, maternal characteristics, household and community-level factors and anemia level was also examined. The study found a significant association between minimum dietary diversity (MDD) and anemia level; emphasizing, that having adequate minimum dietary diversity reduced the likelihood to have higher level of anemia among children aged 6–23 months.

Anemia is still a significant public health issue in sub-Saharan Africa as evidenced by the prevalence of anemia among children aged 6–23 months, which was 72.0% [95% CI: 70.9%, 72.9%] [56]. It is higher than the prevalence reported in earlier research [12,57–61]. One of the possible reason could be due to the long-standing prevalence of malnutrition among children, because of insufficient dietary intake of nutrients, in sub-Saharan Africa [62,63]. In line with previous studies [3,64], the current study revealed that consumption of the adequately diversified diet reduced the risk of anemia among children aged 6–23 months. In addition, due to their frequent exposure to substandard sanitation and environments that encourage the transmission and spread of parasites, children in sub-Saharan Africa are particularly vulnerable to infectious diseases such malaria, hookworms, Schistosoma, and visceral leishmaniasis [65,66].

In the current study, minimum dietary diversity was found to be a protective factor against anemia among children in SSA. Therefore, the study revealed that the diversified diet reduced the risk of anemia. The diversified diet offers children a variety of food groups. These results are consistent with findings reported from other China [3]. Various studies have shown that dietary diversity reduces the risk of malnutrition in infants and children [67,68], and dietary diversity is promoted as an intervention in nutrition in many places [69]. Our study shows that anemia can be reduced by improving the nutritional diversity of complementary foods in SSA. This is supported by the fact that food variety is a good predictor of food quality and micronutrient density in children [70,71]. Therefore, in general, this makes dietary diversity

one of the important things that policy makers should adopt to improve childhood anemia in the country.

Additional, we found in the final model that sex of child, child aged (18–23 months), maternal age, taking drugs for the intestinal parasite in the last six months, maternal education, number of ANC visits, household wealth status, distance of health facility and being born in Central Africa and Southern Africa were significantly associated with the lower odds of severity levels of anemia. Contrarily, child age (9–11 and 12–17 months), size of child at birth, fever in the last two weeks, diarrhea in the last two weeks, birth order, stunting, wasting, and underweight, residence and being in West Africa were significantly associated with higher odds of severity levels of anemia.

In line with previous studies [1,11,12,72], this study revealed that female children were at lower odds of having higher levels of anemia compared to male children. The plausible justification may be male children start complementary feeding too early and the community's perception that male children should receive greater care than female children, which could expose them to various infectious infections and malabsorption issues [73–75]. Compared to children aged 6–8 months, children aged 18–23 months had lower odds of having greater levels of anemia. This is consistent with earlier research that was reported in developing countries [1,72,76], this is due to the fact that prenatal iron storage depletion is evident six months after delivery and puts them at an increased risk of anemia [77].

Being small and average size at birth was significantly associated with increased odds of higher levels of anemia compared to large size at birth. This was in line with studies reported in developing countries [1], India [78] and SSA [12], this could be due to small size at birth being associated with maternal anemia [79], and as a result, children born to anemic mothers may not have enough iron storage [80,81]. As compared to first-order birth, higher birth order was substantially associated with increased likelihood of having higher levels of anemia. This is because greater birth order is associated with substantial maternal nutrition depletion, including iron, folate, and vitamin B12, which could raise the risk of childhood anemia [82]. This is in agreement with study findings in SSA [1,83] and India [84,85]. Furthermore, the competition for food, diseases, and cross contaminations that result from having many children are linked to an increase in socioeconomic and health issues.

Compared to children who did not have diarrhea and a fever, children with a history of those conditions had an increased likelihood of having higher levels of anemia. Studies from Indonesia [86], SSA [12] and Southern Africa [81] that have been reported on are compatible with this. This may be because children with febrile and diarrhoeal illnesses may experience appetite loss and reduced absorption of essential nutrients (iron, folate, and vitamin B12), which may raise the risk of anemia [87]. Additionally, the presence of diarrhea and fever could be a sign of infectious diseases including visceral leishmaniasis, malaria, hookworm, ascariasis, giardiasis, and amoebiasis, which are the main causes of anemia in children [88,89].

This study also revealed that children who took drugs for intestinal parasites in the last 6 months had lower odds of a higher level of anemia compared to children who did not take drugs for intestinal parasites. It is consistent with sub-Saharan Africa [12,90] and Thailand [91]. This may be due to the fact that intestinal parasites can cause anemia, and treating intestinal parasites with medications can reduce the risk of anemia in children [92]. The likelihood of having higher levels of anemia decreased as level of maternal education increased. It is in line with research conducted in Korea [76] and SSA [1,12]. It might be the case since maternal education is a reliable predictor of children's nutritional outcomes [93]. The quality of children's diets is improved by maternal education because it increases the knowledge of mothers regarding newborn nutrition and health (such as exclusive breastfeeding and proper

complementary feeding) [94]. Furthermore, the education level of mothers can have a positive impact on practices relating to child nutrition and health care [95].

Children born to mothers aged 25–34 and 35–49 years had lower odds of higher levels of anemia compared to children of a mother aged 15–24 years. This was consistent with study findings in SSA [1,12]. This may be because mothers who are aged 20 years and older are physiologically mature and have a lower chance of giving birth to low birth weight children than younger women [96]. Children from families with middle and richer household's wealth had decreased odds of having higher levels of anemia than children from poor households. This is in line with study findings in SSA [1,12,83]. This might be the case because children from wealthy families are more likely to be able to feed their children a balanced diet full of macro- and micronutrients, as well as vitamins and minerals [97]. Additionally, children from wealthy families are more likely to have access to medical care for frequent illnesses that cause childhood anemia [98]. Another explanation could be that since poverty is strongly linked to food insecurity, children from low-income households may not have access to foods high in iron, vitamin B12, and folic acid, which in turn raises their chance of developing anemia [99].

In this study, children who were stunted, wasting, and underweight had a higher likelihood of having a higher level of anemia than children who were healthy. This was in line with research studied in Brazil [100] and SSA [12]. Infections and infestations also have synergistic effects of micronutrient deficiencies for causing anemia, in addition to the lack of nutrients needed for hematopoiesis, poor nutritional status is connected with poor immunity [101]. Additionally, children who are undernourished are more likely to have micronutrient deficiencies, such as those in iron, vitamin A, vitamin B12, and folic acid, which are essential for DNA and hemoglobin synthesis during the creation of red blood cells and cause anemia [101].

Children born to mothers who had four or more ANC visits during pregnancy had a decreased risk of having greater levels of anemia than children born to mothers who didn't have ANC visits. This is consistent with study in SSA[1]. This could be because the risk of children anemia is decreased when pregnant women take iron-folate supplements and receive quick diagnosis and treatment for infections including malaria, visceral leishmaniasis, and hookworm, which are known to be the main causes of anemia [102]. Finally, in line with earlier study [1] the odds of having higher levels of anemia among children of mothers perceiving distance to a health facility as not big problem were lower than their counterparts. This might be due to the big distance to the nearby health facility being linked to poor health service utilization for basic services.

## Strengths and limitations of the study

This study was conducted based on a pooled nationally representative DHS survey of the 32 SSA. Additionally, the data was weighted, and a multilevel ordinal logistic regression analysis was done to get a reliable estimate and standard error. Furthermore, this study was based on a large sample size that had adequate power to detect the true effect of the independent variable. Despite the strengths listed above, this study has the following limitations. Since the study was based on cross-sectional nature of DHS, we did not see the seasonal variation and establish causal relationships. Additionally, this study only included children who were alive during the data collection, missing fatalities that might have been caused by consequences from anemia (survivor bias). Also, the limitations of our study include the fact that cross-sectional designs cannot completely eliminate recall errors regarding questionnaire information. Furthermore, because this research was based on secondary data, we were unable to look into all the variables that might be associated with childhood anemia, such as dietary diversity indicators like minimum meal frequency, minimum acceptable diet, parasite infestations (such as malaria, visceral

leishmaniasis, and hookworm), previous hospitalization, and use of nutritional supplements (like vitamin B12 and folate). Finally, since the majority of the children's measured birth weights were not found, variables like a child's birth size, a subjective measurement of a child's birth size, were included in this study. This could result in an overestimation or underestimation of the birth size's influence.

## Conclusions

In conclusion, anemia was a significant public health issue among children aged 6–23 months in sub-Saharan Africa. Minimum dietary diversity, sex of child, child aged, maternal age, taking drugs for the intestinal parasite in the last six months, maternal education, number of ANC visits, household wealth status, size of child at birth, fever in the last two weeks, diarrhea in the last two weeks, birth order, stunting, wasting, underweight, distance of health facility, residence and sub-Saharan African regions were significantly associated with levels of anemia among children aged 6–23 months. Consumption of the minimum dietary diversity reduced the risk of anemia. Reducing anemia in children aged 6–23 months can be achieved by raising mother education levels, treating febrile illnesses, and improve the family's financial situation. Additionally, iron fortification or vitamin supplementation could help to better reduce the risk of anemia and raise children's hemoglobin levels in order to treat anemia. Finally, in order to reduce child anemia, it is preferable to increase the measures for early detection and management of stunted, wasted, and underweight children.

## Supporting information

**S1 Checklist. STROBE statement—Checklist of items that should be included in reports of observational studies.**
(DOCX)

## Author Contributions

**Conceptualization:** Abdu Hailu Shibeshi, Kusse Urmale Mare, Bizunesh Fantahun Kase, Betel Zelalem Wubshet, Tsion Mulat Tebeje, Yordanos Sisay Asgedom, Zufan Alamrie Asmare, Hiwot Altaye Asebe, Afework Alemu Lombebo, Kebede Gemeda Sabo, Bezawit Melak Fente, Beminate Lemma Seifu.

**Data curation:** Abdu Hailu Shibeshi, Beminate Lemma Seifu.

**Formal analysis:** Abdu Hailu Shibeshi, Beminate Lemma Seifu.

**Funding acquisition:** Abdu Hailu Shibeshi.

**Investigation:** Abdu Hailu Shibeshi, Beminate Lemma Seifu.

**Methodology:** Abdu Hailu Shibeshi, Kusse Urmale Mare, Beminate Lemma Seifu.

**Software:** Abdu Hailu Shibeshi, Beminate Lemma Seifu.

**Supervision:** Abdu Hailu Shibeshi, Beminate Lemma Seifu.

**Validation:** Abdu Hailu Shibeshi, Kusse Urmale Mare, Bizunesh Fantahun Kase, Beminate Lemma Seifu.

**Visualization:** Abdu Hailu Shibeshi, Kusse Urmale Mare, Bizunesh Fantahun Kase, Beminate Lemma Seifu.

**Writing – original draft:** Abdu Hailu Shibeshi, Beminate Lemma Seifu.

**Writing – review & editing:** Abdu Hailu Shibeshi, Kusse Urmale Mare, Bizunesh Fantahun Kase, Betel Zelalem Wubshet, Tsion Mulat Tebeje, Yordanos Sisay Asgedom, Zufan Alamrie Asmare, Hiwot Altaye Asebe, Afework Alemu Lombebo, Kebede Gemeda Sabo, Bezawit Melak Fente, Beminate Lemma Seifu.

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
