## [Decision Letter · Decision Letter 0]

31 Oct 2023

PONE-D-23-21819The effect of dietary diversity on anemia levels among children 6-23 months in Sub-Saharan Africa: A multilevel ordinal logistic regression modelPLOS ONE

Dear Dr. Shibeshi,

Thank you for submitting your manuscript to PLOS ONE. After careful consideration, we feel that it has merit but does not fully meet PLOS ONE’s publication criteria as it currently stands. Therefore, we invite you to submit a revised version of the manuscript that addresses the points raised during the review process.

We look forward to receiving your revised manuscript.

Kind regards,

Sukumar Vellakkal

Academic Editor

PLOS ONE

https://bmcpublichealth.biomedcentral.com/articles/10.1186/s12889-021-10699-8

https://journals.lww.com/jfmpc/Fulltext/2023/02000/Association_of_minimum_dietary_diversity_with.20.aspx

In your revision ensure you cite all your sources (including your own works), and quote or rephrase any duplicated text outside the methods section. Further consideration is dependent on these concerns being addressed

Reviewers' comments:

Reviewer's Responses to Questions

**Comments to the Author**

1. Is the manuscript technically sound, and do the data support the conclusions?

Reviewer #1: Yes

Reviewer #2: Yes

Reviewer #3: Partly

2. Has the statistical analysis been performed appropriately and rigorously? 

Reviewer #1: Yes

Reviewer #2: Yes

Reviewer #3: Yes

3. Have the authors made all data underlying the findings in their manuscript fully available?

Reviewer #1: Yes

Reviewer #2: Yes

Reviewer #3: Yes

4. Is the manuscript presented in an intelligible fashion and written in standard English?

Reviewer #1: Yes

Reviewer #2: Yes

Reviewer #3: No

5. Review Comments to the Author

Reviewer #1: The research was done to understand the relationship between MDD and anemia among children in SSA. This is a nice study with large sample size covering multiple countries in Africa. The author needs to be very specific in focusing the relationship between MDD and anemia.

Reviewer #2: The authors have conducted an analysis of predictors of anemia severity in diverse sub-Saharan African populations using the Demographic and Health Surveys. Overall the manuscript is quite well written. The methods, results and conclusions are described with clarity. The findings are novel. I just have some minor comments.

1. Table 5. I assume the AOR is the proportionate odds ratio. This should be clarified in the title or footnotes.

2. Please provide a description of "cut-points." It is not clear what this is referring to.

3. Page 23. There seem to be a problem with error associated with references. That should be fixable.

4. With regard to "cluster" (page 17), I assume the cluster is based on the Enumeration Area. Please clarify.

Otherwise, the paper adds significant knowledge to this issue.

Reviewer #3: As the reviewer, I have conducted a detailed assessment of the manuscript titled "The effect of dietary diversity on anemia levels among children 6-23 months in Sub-Saharan Africa: A multilevel ordinal logistic regression model" using the STROBE Statement checklist for observational studies. Here is an in-depth report addressing each of the checklist items and providing specific line references where issues have been identified:

Title and Abstract:

1. (a) The title accurately indicates the study's design as a multilevel ordinal logistic regression model.

o No issues identified.

2. (b) The abstract effectively describes what was done but falls short in presenting the study's findings in relation to the research questions. The study aimed to determine the association between dietary diversity and anemia prevalence, but the abstract separates these variables. Consequently, the results fail to demonstrate an association between MDD and reduced anemia prevalence, leading to a potentially misleading conclusion (Lines 27-32).

Introduction:

2. The introduction section presents several issues:

o Line 75 should refer to "g/dL" instead of "g/d."

o Lines 79-82 contain repetitive information regarding the effects of anemia on growth and development, which should be restructured.

o There is a repetitive discussion of the causes of anemia in lines 86-91 and 92-95, making it monotonous to read.

o The introduction inadequately addresses the role of dietary diversity in anemia prevalence and lacks relevant literature to establish the rationale.

o Line 111-122 is better suited for the discussion section, as its placement in the introduction is inappropriate. Overall, the introduction is poorly written.

Objectives:

3. The manuscript does not clearly state specific study objectives, which is a critical omission. Specific objectives should be explicitly defined, including any prespecified hypotheses.

Methods:

4. The manuscript accurately presents the key elements of the study design, appropriately considering it as a secondary data analysis.

5. The description of the setting, locations, and data collection dates is correctly provided.

6. Since this is a secondary data analysis, the absence of eligibility criteria and participant selection details is acceptable.

7. All dependent and predictor variables are well-defined.

8. While the sources of data for each variable are described, the method of data collection for minimum dietary diversity is not mentioned.

9. The manuscript describes efforts to address potential sources of bias, considering various variables that may influence anemia levels in children.

10. No sample size calculation is provided, but the distribution of sample size by survey is tabulated. It remains unclear if the reported 52,180 was the total sample for the 32 countries. This issue should be clarified to ensure the accuracy of sample size representation.

11. The manuscript explains how quantitative variables were handled in the analyses, including the use of different analytical techniques. However, irrelevant model types are described (Lines 209-215), which should be removed. Ethical considerations should be moved before the method section, following the study setting for better organization.

12. The statistical methods, including multilevel logistic regression to control for potential confounding variables, are well-described. However, it is not indicated how missing data were analyzed.

Results:

13. There are several issues in the results section:

• Discrepancies in participant numbers for different countries and missing data (Line 278, Table 2).

• There is discrepancies such as dietary diversity data being available for 746 individuals in Swaziland instead of 763, in Gambia 3527 instead of 1172, and in South Africa 867 in instead of 293. This is just few discrepancies as an example. Thus, the manuscript should clarify how the sample size for each country was determined and address the potential bias introduced by these variations.

• The reference error "Error ! Reference source not found" needs clarification.

• Missing data in baseline variables should be explained, along with the methods used to address the missing data.

• Lines 304 to 308 are better placed in the discussion section.

• Table 4 should include anemia vs. dietary diversity, and a subgroup analysis for different categories of anemia. It is recommended to perform a logistic regression model for anemia vs. MDD, adjusting for various socioeconomic confounders instead of relying on the chi-square results (Table 4).

• The manuscript should clarify the model fit statistics referenced in the text (Lines 313-323). It is not clear why different anemia levels are being compared.

Discussion:

18. The discussion should be refocused on the primary study objective, which is the association between anemia levels and MDD. The manuscript appears to emphasize individual significant results of secondary predictors instead.

• While the manuscript mentions the large sample size as a strength indicating there is adequate power to detect the true effect of the independent variable. However, it does not provide a power calculation to support this claim.

• Some limitations are discussed, but the limitation related to the 24-hour recall for minimum dietary diversity is not mentioned and there is potential social desirability bias that could impact on the results. Discuss finding should be interpreted with caution.

Recommendation

This comprehensive report outlines the issues identified throughout the manuscript, with specific line references and requires substantial revisions to address these concerns and enhance the clarity, relevance, and presentation of results. After making the necessary revisions, a reevaluation is recommended for potential publication.

6. PLOS authors have the option to publish the peer review history of their article (what does this mean?). If published, this will include your full peer review and any attached files.

Reviewer #1: **Yes: **Md Kamruzzaman

Reviewer #2: No

Reviewer #3: No

---

## [Author Response · Author response to Decision Letter 0]

28 Nov 2023

Reviewer 1: Comments with response

Thank you for spending your golden time and providing constructive comments on this manuscript

Abstract:

Line 61: Though child under-nutrition like stunting, wasting, and underweight is crucial to address early, these are not the focal point of this research. Therefore, in my opinion it is not necessary to mention this here. 

 Thank you for this important comment related to the reality, then by considering this comment we have adjust while preparing revised manuscript

Introduction:

Line 88: ID need to be abbreviated at first.

 Thank you for this important comment, then by considering this comment we have incorporate it while preparing revised manuscript

Methodology: 

Line 209: It was also better to perform binary logistic regression analysis converting anaemia as a binary variable (anaemic vs non-anaemic)

 Thanks for this comment related to the nature of variable. The outcome variable (anaemia) can be categorized as anaemia (anaemic vs non-anaemic) or anaemia level (non-anaemic, mild, moderate, and severe anaemic). Even though one can perform binary logistic regression analysis converting anaemia as binary variable (anaemic vs non-anaemic), most of scholars or evidence suggest the nature of outcome variable (anaemic level) is ordinal. So in line with ranked nature of the outcome variable, we have perform ordinal logistic regression analysis converting anaemia level as ordinal (non-anaemic, mild, moderate, and severe anaemic).

Results:

Line 277: Reference need to be corrected.

 Thank you for your comment. We have incorporated this comment while preparing revised manuscript.

Line 282: Reference need to be corrected.

 Thank you for your comment. We have incorporated this comment while preparing revised manuscript.

Discussion:

The articles focal point is the relation between MDD and anaemia among children. However, in the discussion part little has been discussed about this. Other secondary factors have been discussed in detail. In my opinion, the author should need to focus on MDD and anaemia, rather than on other factors.

 Thank you for this significant comment related to reality, by taking this as a comment we have included this assessment on the revised manuscript after carefully referring the result.

General Comments:

The research was done to understand the relationship between MDD and anaemia among children in SSA. This is a nice study with large sample size covering multiple countries in Africa. The author needs to be very specific in focusing the relationship between MDD and anaemia.

Reviewer 2: Comments and response

Reviewer #2: The authors have conducted an analysis of predictors of anemia severity in diverse sub-Saharan African populations using the Demographic and Health Surveys. Overall the manuscript is quite well written. The methods, results and conclusions are described with clarity. The findings are novel. I just have some minor comments.

1. Table 5. I assume the AOR is the proportionate odds ratio. This should be clarified in the title or footnotes.

2. Please provide a description of "cut-points." It is not clear what this is referring to.

3. Page 23. There seem to be a problem with error associated with references. That should be fixable.

4. With regard to "cluster" (page 17), I assume the cluster is based on the Enumeration Area. Please clarify.

Otherwise, the paper adds significant knowledge to this issue.

Response: #1, Thank you for these comments. By considering this comment we have incorporates while preparing the revised manuscript.

Response: #2, Cut-points used to obtained the odd ratio results. The Brant test of parallel odds assumption showed that odds ratios appeared to have held constant across all cut-off points of childhood anemia status for the final model at a 5% significance level (p-value = 0.08). The interpretations of odds ratio results obtained by modeling severely anemic vs moderately/mild/non-anemic; and anemic vs non-anemic were the same.

Response: #3, Thanks for this comment and by considering this we have fixed while preparing revised manuscript.

Response: #4, Yes you are right and we have mentioned under data source and study setting as well.

Reviewer 3: Comments and response

Thanks for spending your golden time and providing constructive comments on this manuscript. 

Title and Abstract:

 (a) The title accurately indicates the study's design as a multilevel ordinal logistic regression model.

 No issues identified.

 (b) The abstract effectively describes what was done but falls short in presenting the study's findings in relation to the research questions. The study aimed to determine the association between dietary diversity and anemia prevalence, but the abstract separates these variables. Consequently, the results fail to demonstrate an association between MDD and reduced anemia prevalence, leading to a potentially misleading conclusion (Lines 27-32).

Response: Thanks for this helpful comment related to the reality, and then by considering this comment we have incorporated while preparing revised manuscript.

Introduction:

 The introduction section presents several issues:

 Line 75 should refer to "g/dL" instead of "g/d."

 Lines 79-82 contain repetitive information regarding the effects of anemia on growth and development, which should be restructured.

 There is a repetitive discussion of the causes of anemia in lines 86-91 and 92-95, making it monotonous to read.

 The introduction inadequately addresses the role of dietary diversity in anemia prevalence and lacks relevant literature to establish the rationale.

 Line 111-122 is better suited for the discussion section, as its placement in the introduction is inappropriate. Overall, the introduction is poorly written.

Response: Thank you very much for this constructive comments related to the issues under introduction. As you know under introduction, it is better to show the efforts made by the concerned body to address the issue and the gap we have tried to fill by conducting this study. So by considering this fact, we have incorporated all comments while preparing the revised manuscript.

Objectives:

 The manuscript does not clearly state specific study objectives, which is a critical omission. Specific objectives should be explicitly defined, including any prespecified hypotheses.

Response: Thanks for this comment. But every journals have own “manuscript body formatting guidelines”. So by considering the “manuscript body formatting guidelines” of “PLOS ONE” we have stated the objective under background. 

Methods:

 The manuscript accurately presents the key elements of the study design, appropriately considering it as a secondary data analysis.

 The description of the setting, locations, and data collection dates is correctly provided.

 Since this is a secondary data analysis, the absence of eligibility criteria and participant selection details is acceptable.

 All dependent and predictor variables are well-defined.

 While the sources of data for each variable are described, the method of data collection for minimum dietary diversity is not mentioned.

 The manuscript describes efforts to address potential sources of bias, considering various variables that may influence anemia levels in children.

 No sample size calculation is provided, but the distribution of sample size by survey is tabulated. It remains unclear if the reported 52,180 was the total sample for the 32 countries. This issue should be clarified to ensure the accuracy of sample size representation.

 The manuscript explains how quantitative variables were handled in the analyses, including the use of different analytical techniques. However, irrelevant model types are described (Lines 209-215), which should be removed. Ethical considerations should be moved before the method section, following the study setting for better organization.

 The statistical methods, including multilevel logistic regression to control for potential confounding variables, are well-described. However, it is not indicated how missing data were analyzed.

Response: Since the study was based on the most recent Demographic and Health Survey (DHS) data of 32 SSA countries. By considering this fact, we have incorporated while preparing the revised manuscript as follows:

 No (8): Regarding minimum dietary diversity: When children aged 6–23 months receive four or more of the seven dietary groups, it is said to have MDD [48-50]. Along with breast milk, the children were required to eat from at least four of the seven food categories. The seven food classes were: cereals, roots, and tubers; legumes and nuts; dairy products (milk, yogurt, cheese); flesh foods (meat, fish, fowl, liver, or other organs); eggs; fruits and vegetables high in vitamin A; and other fruits and vegetables. The scores for each food group, which ranged from 0 to 7, were added up to estimate MDD. A score of one (1) was given to any child who consumed any of the food groups, and a score of zero (0) was given to any child who did not. Children who consumed at least four (≥4) of the food groups were considered to have an adequate MDD, and this was coded as "1", whereas the remaining children who consumed fewer than four food groups were coded as "0 = inadequate". This classification and categorization was inspired by studies on either dietary diversity by itself or its association to under-nutrition [48-50]

No (10): Since the source of data is secondary data and then we extracted the data from the KR dataset based on literature and then appended using the STATA command “append using”. A sample of 52,180 mother-child pairs of children within the ages of 6–23 months and mothers aged 15–49 years were included in this study (Table 1)

No (11): We have incorporated this comment while preparing the revised manuscript.

No (12): Thank you this comment related to science. Generally, one can conduct the outcome with missing data by including in the analysis through multiple imputations to handling missing data on the outcome. But in our study prior to statistical analysis, the data were weighted using sampling weight, primary sampling unit, and strata to restore the survey's representativeness and provide accurate statistical estimates. 

Results:

 There are several issues in the results section:

 Discrepancies in participant numbers for different countries and missing data (Line 278, Table 2).

 There is discrepancies such as dietary diversity data being available for 746 individuals in Swaziland instead of 763, in Gambia 3527 instead of 1172, and in South Africa 867 in instead of 293. This is just few discrepancies as an example. Thus, the manuscript should clarify how the sample size for each country was determined and address the potential bias introduced by these variations.

 The reference error "Error ! Reference source not found" needs clarification.

 Missing data in baseline variables should be explained, along with the methods used to address the missing data.

 Lines 304 to 308 are better placed in the discussion section.

 Table 4 should include anemia vs. dietary diversity, and a subgroup analysis for different categories of anemia. It is recommended to perform a logistic regression model for anemia vs. MDD, adjusting for various socioeconomic confounders instead of relying on the chi-square results (Table 4).

 The manuscript should clarify the model fit statistics referenced in the text (Lines 313-323). It is not clear why different anemia levels are being compared.

Response: Thank you very much for given us several issues under result. Simply the data were weighted using sampling weight, primary sampling unit, and strata to restore the survey's representativeness and provide accurate statistical estimates for each county. All issues raised by reviewer were incorporated while preparing a revised manuscript. The odds ratios for the final model appeared to have maintained constant across all cut-off points for the presence of childhood anemia, according to the Brant test of parallel odds assumption (p-value = 0.08 at a 5% level of significance). As a result, the odds ratio results from the models used to compare anemic to non-anemic and severely anemic to mildly/moderately/non-anemic had the same interpretations.

Discussion:

 The discussion should be refocused on the primary study objective, which is the association between anemia levels and MDD. The manuscript appears to emphasize individual significant results of secondary predictors instead.

 While the manuscript mentions the large sample size as a strength indicating there is adequate power to detect the true effect of the independent variable. However, it does not provide a power calculation to support this claim.

 Some limitations are discussed, but the limitation related to the 24-hour recall for minimum dietary diversity is not mentioned and there is potential social desirability bias that could impact on the results. Discuss finding should be interpreted with caution.

Response: Thanks for those comments related to reality. So by taking those comments we have incorporates the issue while preparing the revised manuscript as well. Also we have mentioned the limitation as ‘Also, the limitations of our study include the fact that cross-sectional designs cannot completely eliminate recall errors regarding questionnaire information’.

---

## [Decision Letter · Decision Letter 1]

16 Jan 2024

PONE-D-23-21819R1The effect of dietary diversity on anemia levels among children 6-23 months in Sub-Saharan Africa: A multilevel ordinal logistic regression modelPLOS ONE

Dear Dr. Shibeshi,

Thank you for submitting your manuscript to PLOS ONE. After careful consideration, we feel that it has merit but does not fully meet PLOS ONE’s publication criteria as it currently stands. Therefore, we invite you to submit a revised version of the manuscript that addresses the points raised during the review process.

We look forward to receiving your revised manuscript.

Kind regards,

Sukumar Vellakkal

Academic Editor

PLOS ONE

Journal Requirements:

Reviewers' comments:

Reviewer's Responses to Questions

**Comments to the Author**

1. If the authors have adequately addressed your comments raised in a previous round of review and you feel that this manuscript is now acceptable for publication, you may indicate that here to bypass the “Comments to the Author” section, enter your conflict of interest statement in the “Confidential to Editor” section, and submit your "Accept" recommendation.

Reviewer #1: All comments have been addressed

Reviewer #2: All comments have been addressed

Reviewer #3: (No Response)

2. Is the manuscript technically sound, and do the data support the conclusions?

Reviewer #1: Yes

Reviewer #2: Yes

Reviewer #3: Yes

3. Has the statistical analysis been performed appropriately and rigorously? 

Reviewer #1: Yes

Reviewer #2: Yes

Reviewer #3: Yes

4. Have the authors made all data underlying the findings in their manuscript fully available?

Reviewer #1: Yes

Reviewer #2: Yes

Reviewer #3: Yes

5. Is the manuscript presented in an intelligible fashion and written in standard English?

Reviewer #1: Yes

Reviewer #2: Yes

Reviewer #3: Yes

6. Review Comments to the Author

Reviewer #1: The author has addressed all comments properly. In my opinion, the English language can be improved a bit mor to remove typos and error.

Reviewer #2: (No Response)

Reviewer #3: Title and Abstract:

• Address minor grammar issues in lines 28 and 29-32.

• Refine lines 27-32 for clarity.

• Considering that many readers may not go through the entire paper, it is crucial to include odds ratios and 95% CI for lines 48-54, ensuring well-reported results in the abstract.

Introduction:

• A reference is needed for lines 99-102.

• Move lines 105-113 to the study strengths in the discussion section.

• Ethical considerations are better suited for inclusion in line 142 and may depend on the specific format requirements of the journal.

Methods:

• Clarify the rationale for using a subset of DHS samples (52,180 participants for 32 countries). Consider using all available data (more than 52,180 participants). Provide a clear rationale for excluding some participants, specifying the criteria used for participant selection to address potential bias.

• Explain how missing data were addressed. Clarify that the analysis was based on a complete case and discuss the implications of this in the strength and weakness section of the discussion. Survey weights are not designed to correct for missing data; rather, they account for the sampling distribution during data collection.

Results:

• Clarify the error "Error! Reference source not found."

• Report only the model with the best fit and present it in your tables. Remove irrelevant results, such as the null model and model 2, from the tables. Instead, focus on reporting model 3 if that is the model with the best fit. This will enhance clarity and reader comprehension.

Recommendation:

• Overall, once these minor adjustments are made, your paper will be ready for publication. Additionally, note that the paper will require editing, and I expect that PLOS ONE will undertake this process once accepted for publication.

7. PLOS authors have the option to publish the peer review history of their article (what does this mean?). If published, this will include your full peer review and any attached files.

Reviewer #1: **Yes: **Md Kamruzzaman

Reviewer #2: No

Reviewer #3: No

---

## [Author Response · Author response to Decision Letter 1]

20 Jan 2024

Reviewer 3: Comments and response

We thank you for spending your golden time and providing constructive comments on this manuscript.

Title and Abstract:

• Address minor grammar issues in lines 28 and 29-32.

Response: Thanks for this helpful comment related to the grammar issue, and then by considering this comment we have incorporated while preparing revised manuscript

• Refine lines 27-32 for clarity.

Response: We have incorporated while preparing revised manuscript

• Considering that many readers may not go through the entire paper, it is crucial to include odds ratios and 95% CI for lines 48-54, ensuring well-reported results in the abstract.

Response: Great thanks for this helpful comment related to well reported results, and then by considering this comment we have included odds ratios with 95%CI and incorporated while preparing revised manuscript

Introduction:

• A reference is needed for lines 99-102.

Response: We thank you for your helpful comment related to reference needed, and then by considering this comment we have cited and incorporated while preparing revised manuscript.

• Move lines 105-113 to the study strengths in the discussion section.

Response: We thank you for your helpful comment, and then by considering this comment we have incorporated while preparing revised manuscript.

• Ethical considerations are better suited for inclusion in line 142 and may depend on the specific format requirements of the journal.

Response: We thank you for your comment, based on the specific format requirements of the journal we included at the end of the method section (lines 250-254) and then by considering this comment we have incorporated while preparing revised manuscript

Methods:

• Clarify the rationale for using a subset of DHS samples (52,180 participants for 32 countries). Consider using all available data (more than 52,180 participants). Provide a clear rationale for excluding some participants, specifying the criteria used for participant selection to address potential bias.

Response: We thank you for your useful comment related to rationality of selecting a sample, and then by considering this comment we have incorporated while preparing revised manuscript (lines 133-139).

• Explain how missing data were addressed. Clarify that the analysis was based on a complete case and discuss the implications of this in the strength and weakness section of the discussion. Survey weights are not designed to correct for missing data; rather, they account for the sampling distribution during data collection.

Response: We thank you for your useful comment related to explain how the missing data was addressed, and then by considering this comment we have incorporated while preparing revised manuscript (lines 511-528).

Results:

• Clarify the error "Error! Reference source not found."

Response: We thank you for your comment. But dear reviewer, we are not seeing any “Error! Reference source not found” under results and if you see any “Error” please let us know where it is specifically and we will fix it. If it is related to figures, we have entered the figures separately as journal guideline. We put only the figures tittle in the document. And again, let us know the (Lines__) that says “Error! Reference source not found”.

• Report only the model with the best fit and present it in your tables. Remove irrelevant results, such as the null model and model 2, from the tables. Instead, focus on reporting model 3 if that is the model with the best fit. This will enhance clarity and reader comprehension.

Response: We thank you for your suggestions and comments. Having this, from our point of view, we have the idea that all models (Null model, Model I, Model II, and Model III) should be present so that we can clearly understand the results of null model (result of rando effect), Model I (result of individual level), Model II (result of community level) and Model III (result of individual and community level jointly). But dear reviewer, if you don’t believe it and you need to change it, we are willing to change it again.

Recommendation:

• Overall, once these minor adjustments are made, your paper will be ready for publication. Additionally, note that the paper will require editing, and I expect that PLOS ONE will undertake this process once accepted for publication.

---

## [Editor Report · Decision Letter 2]

30 Jan 2024

The effect of dietary diversity on anemia levels among children 6-23 months in Sub-Saharan Africa: A multilevel ordinal logistic regression model

PONE-D-23-21819R2

Dear Dr. Shibeshi,

We’re pleased to inform you that your manuscript has been judged scientifically suitable for publication and will be formally accepted for publication once it meets all outstanding technical requirements.

Kind regards,

Sukumar Vellakkal

Academic Editor

PLOS ONE
---

## [Editor Report · Acceptance letter]

10 May 2024

PONE-D-23-21819R2 

PLOS ONE

Dear Dr. Shibeshi, 

I'm pleased to inform you that your manuscript has been deemed suitable for publication in PLOS ONE. Congratulations! Your manuscript is now being handed over to our production team.

Kind regards, 

on behalf of

Dr. Sukumar Vellakkal 

Academic Editor

PLOS ONE